# Calcium Affects Polyphosphate and Lipid Accumulation in Mucoromycota Fungi

**DOI:** 10.3390/jof7040300

**Published:** 2021-04-15

**Authors:** Simona Dzurendova, Boris Zimmermann, Achim Kohler, Kasper Reitzel, Ulla Gro Nielsen, Benjamin Xavier Dupuy--Galet, Shaun Leivers, Svein Jarle Horn, Volha Shapaval

**Affiliations:** 1Faculty of Science and Technology, Norwegian University of Life Sciences, Drøbakveien 31, 1433 Ås, Norway; boris.zimmermann@nmbu.no (B.Z.); achim.kohler@nmbu.no (A.K.); benjamin.dupuy.galet@nmbu.no (B.X.D.--G.); volha.shapaval@nmbu.no (V.S.); 2Department of Biology, University of Southern Denmark, Campusvej 55, DK-5230 Odense M, Denmark; reitzel@biology.sdu.dk; 3Department of Physics, Chemistry and Pharmacy, University of Southern Denmark, Campusvej 55, DK-5230 Odense M, Denmark; ugn@sdu.dk; 4Faculty of Chemistry, Biotechnology and Food Science, Norwegian University of Life Sciences, Christian Magnus Falsens vei 1, 1433 Ås, Norway; shaun.allan.leivers@nmbu.no (S.L.); svein.horn@nmbu.no (S.J.H.)

**Keywords:** Mucoromycota, calcium, lipids, polyphosphates, carotenoids, biorefinery, fungi

## Abstract

Calcium controls important processes in fungal metabolism, such as hyphae growth, cell wall synthesis, and stress tolerance. Recently, it was reported that calcium affects polyphosphate and lipid accumulation in fungi. The purpose of this study was to assess the effect of calcium on the accumulation of lipids and polyphosphate for six oleaginous Mucoromycota fungi grown under different phosphorus/pH conditions. A Duetz microtiter plate system (Duetz MTPS) was used for the cultivation. The compositional profile of the microbial biomass was recorded using Fourier-transform infrared spectroscopy, the high throughput screening extension (FTIR-HTS). Lipid content and fatty acid profiles were determined using gas chromatography (GC). Cellular phosphorus was determined using assay-based UV-Vis spectroscopy, and accumulated phosphates were characterized using solid-state ^31^P nuclear magnetic resonance spectroscopy. Glucose consumption was estimated by FTIR-attenuated total reflection (FTIR-ATR). Overall, the data indicated that calcium availability enhances polyphosphate accumulation in Mucoromycota fungi, while calcium deficiency increases lipid production, especially under acidic conditions (pH 2–3) caused by the phosphorus limitation. In addition, it was observed that under acidic conditions, calcium deficiency leads to increase in carotenoid production. It can be concluded that calcium availability can be used as an optimization parameter in fungal fermentation processes to enhance the production of lipids or polyphosphates.

## 1. Introduction

Mucoromycota fungi are powerful cell factories widely applicable in developing modern biorefineries [1,2]. Mucoromycota fungi can accumulate a wide range of high-value metabolites, among which lipids and polyphosphates have gained much interest in recent years [3]. Oleaginous Mucoromycota can accumulate lipids with a similar composition as plant and animal oils, in amounts higher than 20% of their dry cell biomass [4].

In order to optimize the production of Mucoromycota lipids and polyphosphate and maximize biomass yield, it is crucial to understand the role of different growth medium components on fungal growth and metabolic activity [5,6]. In a recent study, we investigated the effect of metal and phosphate ions on the growth, lipid accumulation, and cell chemistry of *Mucor circinelloides* [7]. We showed that calcium (Ca) starvation enhanced lipid accumulation in *M. circinelloides,* while increased Ca availability positively affected polyphosphate accumulation.

Calcium is a unique universal signaling element in prokaryotic and eukaryotic cells. Calcium signaling is an evolutionary conserved process, which, in fungal cells, regulates multiple cell functions ranging from growth [8,9,10], hyphae development, sporulation, and chitin synthesis [11] to intracellular pH signaling [12], stress tolerance, and virulence [13]. The level of Ca^2+^ in the cytosol is important for signaling and regulation of the above-mentioned processes. In fungal cells, calcium is mainly stored in vacuoles, which can contain approximately 95% of the cellular Ca [14]. For supporting Ca signaling, cells maintain cytosolic Ca at a low concentration. There are different protein transporters managing the level of Ca ions in cytosol and mediating entry or exit from vacuoles. In eukaryotic cells, Ca is required at the endoplasmic reticulum (ER), where it provides the correct function of protein folding and secretory machinery [15].

Since polyphosphate and lipid accumulation are associated with ER, calcium could be directly or indirectly involved in their accumulation. It has been reported that calcium and several other cations neutralize the negative charge of polyphosphate in fungal cells [16,17]. Thus, it can be hypothesized that with a higher availability of calcium ions in the medium, more efficient neutralization of the negatively charged polyphosphate occurs and, subsequently, a higher amount of phosphorus can be stored intracellularly in the form of polyphosphate [7]. Moreover, it has been reported that calcium starvation enhances lipid accumulation in oleaginous algae [18] and mammalian adipocyte cells [19]. Currently, there are several hypotheses on the mechanisms behind Ca-deficiency-induced lipid accumulation in oleaginous microorganisms. The first hypothesis is related to the study by Cifuentes et al. [20]. It is based on mediation of antilipolytic pathways through a calcium-sensing receptor (CaSR) triggered by the low cellular availability of Ca ions. This results in enhanced lipid accumulation in cells. Due to the evolutionary conservation of lipolytic pathways and Ca signaling [21], it has been suggested that Ca deficiency can mediate similar antilipolytic pathways in oleaginous microorganisms [7]. The second hypothesis was suggested by Wang et al. [19], and it is based on the importance of calcium ions in the basal sensitivity of the sterol-sensing mechanism of the sterol response element binding protein (SREBP) pathway. Wang et al. discovered that a reduction in the Ca concentration in ER changes the distribution of intracellular sterol/cholesterol, resulting in the enhancement of SREBP activation and triggering the synthesis of neutral lipids. Sterol response element binding proteins (SREBPs) are transcription factors that are synthesized on ER and are considered as ER-associated integral membrane proteins [19]. SREBPs were reported for eukaryotic cells, including mammalian and fungal cells [22].

In order to investigate whether the role of calcium ions in lipid and polyphosphate accumulation is conserved for different Mucoromycota fungi, in this study, six Mucoromycota strains were grown in the presence or absence of Ca ions at three different phosphorus concentrations and, thus, different pH conditions. Relatively high phosphate concentrations were used to buffer the growth media and provide conditions for polyphosphate accumulation. Nitrogen limitation was used to trigger the lipid accumulation in oleaginous fungi.

To the authors’ knowledge, this study is among the first studies assessing the role of Ca ions on polyphosphate and lipid accumulation in Mucoromycota fungi under different phosphorus concentrations.

## 2. Materials and Methods

### 2.1. Fungal Strains

Six oleaginous Mucoromycota fungi from the genera *Amylomyces*, *Mucor*, *Rhizopus*, and *Umbelopsis*, obtained from the Czech Collection of Microorganisms (CCM; Brno, Czech Republic), Norwegian School of Veterinary Science (VI; Ås, Norway), Food Fungal Culture Collection (FRR; North Ryde, Australia), and Université de Bretagne Occidentale Culture Collection (UBOCC; Brest, France), were used in the study (Table 1). The selection of fungal strains was based on the results of our previous studies, where a set of Mucoromycota fungi were examined for the co-production of lipids, chitin/chitosan, and polyphosphates [5,6,23].

### 2.2. Growth Media and Cultivation Conditions

Cultivation media were formulated by implementing a full factorial design, where three different concentrations of inorganic phosphorus substrate (Pi)—phosphate salts KH_2_PO_4_ and Na_2_HPO_4_—and two Ca conditions—Ca1 (presence) and Ca0 (absence)—were used. The cultivation was performed in a Duetz microtiter plate system (MTPS) [24] in four independent biological replicates for each fungus and condition, resulting in 144 samples. Cultivation was carried out in two steps: (1) growth on a standard agar medium for preparing spore inoculum, and (2) growth in a Duetz MTPS in nitrogen-limited broth media with ammonium sulphate as the nitrogen source and different concentrations of Pi and Ca.

For preparation of the spore inoculum, all strains except *U. vinacea* were cultivated on malt extract agar (MEA). *U. vinacea* was cultivated on potato dextrose agar (PDA). MEA was prepared by dissolving 30 g of malt extract agar (Merck, Germany) in 1 L of distilled water and autoclaving at 115 °C for 15 min. PDA was prepared by dissolving 39 g of potato dextrose agar (VWR, Belgium) in 1 L of distilled water and autoclaving at 115 °C for 15 min. Agar cultivation was performed for 7 days at 25 °C for all strains. Fungal spores were harvested from agar plates with a bacteriological loop after the addition of 10 mL of sterile 0.9% NaCl solution.

The main components of the nitrogen-limited broth media [25] with modifications [26] (g/L) were glucose, 80; (NH_4_)_2_SO_4_, 1.5; MgSO_4_·7H_2_O, 1.5; CaCl_2_·2H_2_O, 0.1; FeCl_3_·6H_2_O, 0.008; ZnSO_4_·7H_2_O, 0.001; CoSO_4_·7H_2_O, 0.0001; CuSO_4_·5H_2_O, 0.0001; MnSO_4_·5H_2_O, 0.0001. Chemicals were purchased from Merck (Germany). The concentrations of the phosphate salts, 7 g/L KH_2_PO_4_ and 2 g/L Na_2_HPO_4_, were selected as a reference value (Pi1) since they have frequently been used for cultivation of oleaginous Mucoromycota [25,26]. The broth media contained, in addition to Pi1, a higher Pi4 (4 × Pi1) and a lower Pi0.5 (0.5 × Pi1) amount of phosphate salts. Cultivation in broth media was performed in the Duetz MTPS (Enzyscreen, The Netherlands), which consists of 24-square polypropylene deepwell microtiter plates, low-evaporation sandwich covers, and extra-high cover clamps. The autoclaved microtiter plates were filled with 7 mL of sterile broth medium per well, and each well was inoculated with 50 µL of spore inoculum. The Duetz MTPS were placed into a MAXQ 4000 shaker (Thermo Scientific), and cultivation was performed for 7 days at 25 °C and 400 rpm agitation (1.9 cm circular orbit).

### 2.3. Fourier-Transform Infrared Spectroscopy

#### 2.3.1. FTIR-HTS of Fungal Biomass

Fourier-transform infrared (FTIR) spectroscopy analysis of fungal biomass was performed according to Kosa et al. [26] with some modifications [5]. The biomass was separated from the growth media by centrifugation and washed with distilled water. Approximately 5 mg of fresh washed biomass was transferred into a 2-milliliter polypropylene tube containing 250 ± 30 mg of acid-washed glass beads and 0.5 mL of distilled water for further homogenization. The remaining washed biomass was freeze-dried for 24 h for determining biomass yield.

The homogenization of fungal biomass was performed using a Percellys Evolution tissue homogenizer (Bertin Technologies, France) with the following set-up: 5500 rpm, 6 × 20 s cycle. Briefly, 10 µL of homogenized fungal biomass was pipetted onto an IR transparent 384-well silica microplate. Each biomass sample was analyzed in 3 technical replicates. Samples were dried at room temperature for 2 h.

FTIR spectra were recorded in transmission mode using a high-throughput screening extension (HTS-XT) unit coupled to a Vertex 70 FTIR spectrometer (both Bruker Optik GmbH, Leipzig, Germany). Spectra were recorded in the region between 4000 and 500 cm^−1^, with a spectral resolution of 6 cm^−1^, a digital spacing of 1.928 cm^−1^, and an aperture of 5 mm. For each spectrum, 64 scans were averaged. Spectra were recorded as the ratio of the sample spectrum to the spectrum of the empty IR transparent microplate. In total, 432 biomass spectra were obtained. The OPUS software (Bruker Optik GmbH, Leipzig, Germany) was used for data acquisition and instrument control.

#### 2.3.2. FTIR-ATR of Culture Supernatant

Briefly, 10 µL of culture supernatant was deposited on an attenuated total reflection (ATR) crystal. FTIR reflectance spectra were measured with a single reflectance-attenuated total reflectance (SR-ATR) accessory, high-temperature Golden Gate ATR Mk II (Specac, Orpington, UK), coupled to the Vertex 70 FTIR spectrometer (Bruker Optik GmbH, Leipzig, Germany). The FTIR-ATR spectra were recorded with a total of 32 scans, a spectral resolution of 4 cm^−1^, and a digital spacing of 1.928 cm^−1^, over the range of 4000–600 cm^−1^, using the horizontal SR-ATR diamond prism with a 45° angle of incidence. All samples were analyzed in three technical replicates, and background measurement of the empty crystal was conducted between each sample measurement. The OPUS software (Bruker Optik GmbH, Leipzig, Germany) was used for data acquisition and instrument control.

### 2.4. Analysis of Cellular Phosphorus

#### 2.4.1. Analysis of Total P in Fungal Biomass

Total P was estimated using assay-based UV–Vis spectrometry. Biomass samples were freeze-dried and decomposed in a muffle oven at 550 °C for 16 h. Five milliliters of 6 M HCl was added to each sample. Samples were boiled on a heating plate for 20 min, 7.5 mL MilliQ water was added, and samples were left overnight in the acid/water mixture. The next day, samples were diluted up to 100 mL with MilliQ water, centrifuged, and analyzed using RX Daytona+ with kit PH8328 (Randox) [27].

#### 2.4.2. Solid-State NMR (SSNMR) Characterization of Phosphates in Fungal Biomass

Quantitative ^31^P SSNMR spectra were recorded on a 500-megahertz (MHz) JEOL ECZ 500R spectrometer using a 3.2-millimeter triple resonance magic-angle-spinning (MAS) NMR probe, a 200 kilohertz (kHz) spectral window, 15 kHz spinning speed, and a 45° pulse. Relaxation delays were optimized on each sample, typically 60–90 s for the fungal samples and 250 s for struvite (NH_4_MgPO_4_ 6H_2_O), which served as an external intensity reference for spin counting experiments [28]. The ^31^P SSNMR spectra were referenced relative to 85% H_3_PO_4_ (δ(^31^P) = 0 ppm) and analyzed using MestReNova (Mestrelab Research) by absolute integration of the spinning side band manifold.

### 2.5. Lipid Extraction and GC-FID Analysis of Fatty Acid Profile

Direct transesterification was performed according to Lewis et al. [29] with modifications [30]. Briefly, a 2-milliliter screw-cap polypropylene tube was filled with 20 ± 5 mg freeze-dried biomass, approx. 250 ± 30 mg (710–1180 μm diameter) acid-washed glass beads, and 500 μL of chloroform. Then, 1.05 mg of glyceryl tritridecanoate (C13:0 triacylglycerol- TAG) internal standard in 100 µL of hexane was added to the polypropylene tube. The fungal biomass was homogenized in a Percellys Evolution tissue homogenizer at 5500 rpm for 6 × 20-s cycles. The processed biomass was transferred into a glass reaction tube by washing the polypropylene tube with 2400 μL of methanol–chloroform–hydrochloric acid solvent mixture (7.6:1:1 *v/v*) (3 × 800 µL). Finally, 500 μL of methanol was added into a glass reaction tube. The reaction mixture was incubated at 90 °C for 90 min in a heating block, followed by cooling to room temperature. Then, 1 mL of distilled water was added to the glass reaction tube. Fatty acid methyl esters (FAMEs) were extracted by the addition of 2 mL hexane followed by 10 s vortex mixing. The reaction tube was centrifuged at 3000 rpm for 5 min at 4 °C, and the upper (organic) phase was collected in a glass tube. The lower (aqueous) phase was extracted twice more, this time by the addition of 2 mL hexane–chloroform mixture (4:1 *v/v*). The solvent in the glass tube was evaporated under nitrogen at 30 °C, and a small amount of anhydrous sodium sulphate (approx. 5 mg) was added in the glass tube. FAMEs were transferred into a GC vial by washing the glass tube with 1500 µL hexane (2 × 750 µL) containing 0.01% butylated hydroxytoluene (BHT, Sigma-Aldrich, St. Louis, MO, USA) followed by 5 s vortex mixing. Lipid contents and fatty acid profiles were analyzed using a gas chromatography 7820A System (Agilent Technologies, Santa Clara, CA, USA) as described previously [30].

### 2.6. Data Analysis

The following software packages were used for the data analysis: Unscrambler X version 11 (CAMO Analytics, Oslo, Norway) and Orange data mining toolbox version 3.20 (University of Ljubljana, Slovenia) [31].

#### 2.6.1. Analysis of FTIR Spectral Data of Fungal Biomass

FTIR-HTS spectra of fungal biomass were pre-processed using extended multiplicative signal correction (EMSC) with linear, quadratic, and cubic terms. The amide I peak (1650 cm^−1^), mainly related to proteins, was selected as a relatively stable reference band, used for estimating the relative content of cellular lipids and phosphates. Furthermore, an ester bond in the FTIR spectra (C=O stretching at 1745 cm^−1^) was selected for the estimation of lipids and a phosphate functional group bond (P=O stretching at 1251 cm^−1^) was selected for the estimation of polyphosphates. Thus, lipid-to-protein (LP) and polyphosphate-to-protein (PP) ratios could be effectively estimated.

#### 2.6.2. FTIR-ATR Spectral Data of Culture Supernatants—Glucose Estimation

The residual glucose in the culture supernatants was estimated from the FTIR-ATR spectra of the culture supernatants using a prediction model based on standard solutions with known glucose and phosphates concentrations. All FTIR-ATR spectra were pre-processed by selecting the region of interest, 900–700 cm^−1^; the baseline was corrected using vertical offset; and peaks were normalized using the water band at 1637 cm^−1^. The reference solutions included glucose concentrations from 10 to 100 g/L (in steps of 10 g/L), each with 4 different Pi source levels (Pi4, Pi2, Pi1, and Pi05) [5]. The dataset used for building the partial least square regression (PLSR) model was divided into two subsets: a calibration subset (70%) and a validation subset (30%). The validation subset contained samples with glucose concentrations of 20, 50, and 70 g/L and all Pi concentrations. The model was further externally validated with an additional set of reference glucose solutions with known glucose concentrations and culture supernatants from our previous study [32], where the glucose concentrations were estimated using ultra-high-performance liquid chromatography (UHPLC).

## 3. Results

### 3.1. Growth Characteristics of Mucoromycota Fungi

Fungal biomass formation for Mucoromycota fungi grown in the presence or absence of Ca and at different Pi concentrations is shown in Figure 1, and pH data measured at the end of cultivations are presented in Appendix A. The biomass concentration data represent the average of four bio-replicates, and the error bars show the standard deviation. Generally, the biomass formation was reproducible within the bio-replicates. The initial pH of the culture media was 6.0 ± 0.3, and it stayed relatively stable in Ca-Pi4 conditions, while a drop in the pH down to 3.0 and 2.0 in Pi1 and Pi0.5 conditions, respectively, was observed. Overall, for the studied Mucoromycota, the highest biomass concentration was achieved for *U. vinacea,* which produced from 20.56 to 24.49 g/L of biomass. Fungi from the *Mucor* and *Amylomyces* genera produced biomass in the range of 7.52–12.00 g/L in Ca-Pi4 and Ca-Pi1 media and 5.41–8.70 g/L in Ca-Pi0.5 conditions. The lowest biomass concentration was observed for *R. stolonifer* (Figure 1).

Fungi grown in non-acidic conditions when the pH was relatively stable (Pi4 media) reached higher biomass concentrations than in the media with acidic pH and lower phosphorus availability. This observation was valid for all tested fungi except *R. stolonifer*, which showed relatively similar biomass production from 4.11 to 6.10 g/L for all conditions (Figure 1). The lowest biomass production for the studied Mucoromycota was observed in Pi0.5 media when the pH was acidic. Figure 1 shows that calcium ion deprivation (Ca0) combined with both high (Pi4) and reference (Pi1) amounts of phosphorus led to a slight decrease in the biomass production for all fungi. This is contrary to the fungal growth in the Pi0.5 media, where the biomass production was improved for the Ca0 condition, except for *M. circinelloides* FRR 5020. The most significant effect of calcium ion deficiency on the biomass production was recorded for *U. vinacea,* where the Ca1 and Ca0 conditions yielded a difference of approximately 4 g/L in biomass concentration. Interestingly, the *M. circinelloides* strains reacted differently to the absence of Ca ions in acidic Pi0.5 media. *M. circinelloides* VI 04473 (MC1) formed significantly more biomass in the Ca0-Pi0.5 (8.70 g/L) medium than in Ca1-Pi0.5 (6.24 g/L), while *M. circinelloides* FRR 5020 (MC2) did not show any significant differences in biomass production between Ca1 (6.97 g/L) and Ca0 (6.10 g/L) conditions. Calcium availability had no significant effect on the fungal strains *A. rouxii* and *R. stolonifer* in Pi0.5 media.

The residual glucose, which was not consumed by fungi, was estimated using a regression model based on the FTIR-ATR spectra of culture supernatants and is shown in Figure 2.

Generally, the glucose consumption corresponded well to the biomass production data. The lowest residual glucose was recorded for *U. vinacea* in Ca0-Pi4 media, which correlated well with the highest biomass production detected for this condition (Figure 2). The strain *R. stolonifer* showed the lowest glucose consumption as well as the lowest biomass formation. Interestingly, *R. stolonifer* grown in Pi4 media and in the presence of Ca ions consumed more glucose, and a lower pH was detected in the supernatants than for conditions where Ca was absent. This may be due to the formation of organic acid and the lowering of the pH, since the *Rhizopus* genus is known for extracellular acid production. The strain *A. rouxii* and both *M. circinelloides* strains grown in Pi4 media showed similar glucose consumption, with approximately half of the initial glucose remaining (40 g/L) and reaching a biomass concentration of 10–12 g/L. In contrast, when grown in Pi0.5 media, these fungi consumed approximately 20 g/L of glucose for the production of 5–8 g/L biomass. While *M. racemosus* biomass concentration was similar to that of *A. rouxii* in Pi4 and Pi1 media, significantly less glucose was consumed by *M. racemosus* (Figure 2).

### 3.2. Importance of Ca Ions’ Availability for Polyphosphate Accumulation in Mucoromycota

For studying the influence of Ca ions on the lipid and polyphosphate accumulation in Mucoromycota fungi, we performed FTIR spectroscopic profiling of the composition of the fungal biomass. Total phosphorus was estimated using assay-based UV–Vis spectroscopy, and phosphates were characterized using solid-state ^31^P MAS NMR spectroscopy. Lipid content and fatty acid profiles were assessed using GC system with flame ionization detector (GC-FID).

When examining the FTIR-HTS spectra of fungal biomass grown under different calcium ions conditions, substantial changes were observed for bands associated with lipids (=C-H stretching at 3010 cm^−1^, -C-H stretching at 2920 and 2850 cm^−1^, and -C=O stretching at 1745 cm^−1^) and polyphosphates (P=O stretching 1251 at cm^−1^ and P-O-P stretching at 885 cm^−1^) (Figure 3). An example of such changes is shown with the spectra of *Mucor circinelloides* VI 04473 grown in Ca0-Pi4 and Ca1-Pi4 media (Figure 3).

The analysis of the total phosphorus content in % per cell dry weight (d.w.) of fungal biomass (Figure 4) showed that the highest total phosphorus content was recorded for *A. rouxii* (2.65–6.24%), followed by *M. racemosus* (2.28–5.20%), *M. circinelloides* VI 04473 (1.4–4.91%), and *M. circinelloides* FRR 5020 (1.86–4.50%). *R. stolonifer* and *U. vinacea* showed the most uniform cellular phosphorus content of 2.7–4.13% and 0.64–1.37%, respectively, and the total phosphorus content was the lowest in *U. vinacea* (Figure 4). In the vast majority of samples, it is visible that the phosphorus uptake was enhanced with Ca availability.

In order to confirm that the accumulated phosphorus was stored in the form of polyphosphates, characterization of cellular phosphates was performed using solid-state ^31^P NMR spectroscopy (SSNMR) (Figure 5). Due to the fact that SSNMR is expensive and time consuming, one representative strain, *M. circinelloides* VI 04473, was selected for this analysis. SSNMR showed that *M. circinelloides* VI 04473, under all tested Ca-Pi conditions, contained mainly polyphosphates (more than 90%). Meanwhile, for biomass obtained from the media with Pi4 and Pi05, more polyphosphates were present in the Ca1 condition (97–98%), and for the biomass from Pi1 media, the amount of orthophosphates was comparable for the Ca1 and Ca0 conditions (7–8%) (Figure 5).

Nevertheless, polyphosphates have strong signals in FTIR spectra (P=O stretching peak at 1251 cm^−1^ and P-O-P stretching at 885 cm^−1^); therefore, FTIR spectroscopy was utilized for the analysis of fungal phosphates for all samples.

The evaluation of the underlying correlations between Ca ions’ availability and polyphosphate and lipid accumulation, as well as the estimation of the relative content of these metabolites in fungal biomass, was based on peak ratios of lipid/polyphosphate to protein-related bands (Figure 6). Proteins were selected as a relatively stable component of fungal biomass, since under nitrogen-limiting conditions, cell proliferation stops at the end of the exponential growth phase and, therefore, protein content stays consistent in the lipogenesis phase. To estimate the chemical composition of the biomass, the following representative lipid, polyphosphate, and proteins peaks were used: (i) ester bond C=O stretching peak at 1745 cm^−1^ for lipids, (ii) phosphate functional group P=O stretching peak at 1251 cm^−1^ for polyphosphates, and (iii) amide I C=O stretching peak at 1650 cm^−1^ for proteins. The estimated lipid-to-protein (LP) and phosphate-to-protein (PP) ratios are shown in Figure 6.

It can be seen that the effect of Ca ions on lipid and polyphosphate accumulation is pH- and strain-specific. The PP ratio obtained for biomass grown in Ca-Pi4 media was higher than that for other Pi conditions, indicating high polyphosphate accumulation occurring in Mucoromycota fungi under high phosphorus availability. In the media with high phosphorus availability (Pi4), for all strains except *U. vinacea*, a high PP ratio and a higher amount of polyphosphate could be observed in the presence of Ca (Figure 6A). These observations were in accordance with the results obtained from the analysis of the total phosphorus content (Figure 5). When the reference amount of phosphorus (Pi1) was used, the influence of Ca on polyphosphate accumulation was less visible and more strain-specific. Thus, the PP ratio of *A. rouxii*, *M. circinelloides* FRR 5020, *M. racemosus*, *R. stolonifer*, and *U. vinacea* grown in Ca1-Pi1 medium was higher than in the case of Ca0-Pi1 medium, while it was lower for *M. circinelloides* VI 04473 (Figure 6A). Similar results were observed from the total phosphorus content analysis, except for the strains *M. racemosus* and *U. vinacea*, which showed a higher total phosphorus content in the absence of Ca ions (Figure 4). A strain-specific influence of Ca ions was also obtained for the Pi0.5 condition, where a higher PP ratio in the presence of Ca ions was detected for biomass of *A. rouxii*, *M. circinelloides* FRR 5020, and *R. stolonifer* and a lower one was observed for *M. circinelloides* VI 04473, *M. racemosus*, and *U. vinacea.* Interestingly, according to the PP ratio, the lowest polyphosphate accumulation was recorded for the biomass grown in Pi0.5 media. Total phosphorus content in fungal biomass grown in Ca1-Pi0.5 medium was higher than that in Ca1-Pi1 for all tested fungi (Figure 4). Such discrepancies between FTIR-HTS and the reference total phosphorus analysis could be explained by the possible variation in total protein and polysaccharide contents in different Mucoromycota fungi under different pH/Pi conditions that affected the estimation of peak ratios from FTIR spectra. In addition, a possible explanation for this difference is that the FTIR-based analysis shows phosphate content, while the assay-based UV–Vis analysis reports the total P. Furthermore, the biomass grown in Pi0.5 media is poorer in the cellular content of lipids; thus, the results in % show a higher P content. It is important to note that when the total P values in % are normalized to absolute values in g/L, it becomes apparent that the highest phosphate concentrations were achieved in the Pi4 condition (Appendix A, with concentrations in g/L), and comparable results were obtained for Pi1 and Pi0.5 samples. This observation indicates that the majority of phosphorus uptake takes place in the exponential growth phase. To confirm this observation, we investigated the biochemical composition of *M. circinelloides* VI 04473 in time—after 1 and 5 days of growth (Figure 7). As can be seen in Figure 7, the polyphosphate-related peaks in the FTIR-HTS spectra of *M. circinelloides* biomass can already be detected after the first day of cultivation, indicating that polyphosphate accumulation occurred from the start of the exponential growth phase. A further explanation of the lower P content in % for Pi1 than Pi0.5 media is that the biomass grown in Pi1 showed a higher lipid content, as indicated by the lipid-related peaks in the FTIR spectrum (Figure 7).

### 3.3. Ca Ion Deficiency Can Trigger Lipid Accumulation in Mucoromycota

All tested strains exhibited oleaginous properties across all assessed media. Generally, the reference phosphorus amount (Pi1) was the most suitable for lipid accumulation, since the highest lipid content was found under Pi1 conditions for all strains except for *R. stolonifer*. The lipid-to-protein (LP) ratio derived from the FTIR-HTS spectra and the total lipid content (in % per cell dry weight) of fungal biomass grown in the absence of Ca ions were higher than they were in the presence of Ca ions for several fungi and Pi conditions (Figure 6B, Table 2). Thus, the lipid-triggering effect of Ca deprivation was remarkably pronounced in *M. circinelloides* VI 04473 and *R. stolonifer* under all Pi conditions; in *A. rouxii* and *M. circinelloides* FRR5020 in media with Pi4; in *M. circinelloides* FRR5020 and *U. vinacea* in media with Pi1; and in all fungi except *M. circinelloides* FRR5020 in Pi0.5 conditions (Figure 6B, Table 2 and Appendix A). The effect of Ca ions’ availability was, to some extent, strain-specific, where the highest difference in the total lipid content of fungal biomass grown in Ca1 and Ca0 conditions was recorded for *M. circinelloides* VI 04473, *U. vinacea*, and *M. racemosus*, while smaller differences occurred in *A. rouxii* and *R. stolonifer.* Calcium deficiency led to the highest lipid content in all studied Mucoromycota fungi grown in Pi0.5 media.

Among the studied Mucoromycota fungi, *U. vinacea* accumulated the highest lipid amount, ranging from 52% to 84% (*w*/*w*). Interestingly, the least favorable growth condition, Pi05, which resulted in very acidic pH of the growth media, was the most preferable for lipid accumulation in the Ca0 condition for *M. racemosus*.

As was observed both for FTIR and GC analyses, the most pronounced lipid differences related to the Ca availability were seen for the Pi05 condition for *M. circinelloides VI04473* (22.67% Ca1; 48.05% Ca0), *Mucor racemosus* (22.83% Ca1; 39.63% Ca0), and *U. vinacea* (52.36% Ca1; 66.70% Ca0). Moreover, a higher lipid content was recorded for all Pi conditions for *M. circinelloides* VI04473 (Table 2) in Ca-deprived media. The fatty acid (FA) profiles were quite consistent irrespective of the Ca availability (Appendix A). Some minor variation in the FA profiles can be mostly assigned to variation of Pi availability, which resulted in different pH conditions.

## 4. Discussion

Calcium is an important second messenger in the transduction of cellular signals and cell growth under stress conditions. Exposure of fungal cells to environmental stress triggers an immediate response in cytoplasmic calcium levels. This process is fundamental for the survival of eukaryotic cells. Through a variety of calcium signal transduction mechanisms, fungal cells can tolerate numerous environmental changes, including pH stress [33]. There are at least two calcium-based signal transduction pathways regulating the processes necessary for pH adjustment and ion homeostasis in eukaryotic cells [33]. In this study, fungi were grown in media with ammonium sulphate as a nitrogen source and various phosphorus substrate concentrations combined with different levels of calcium availability. Due to the low buffering capacity of ammonium sulphate and the fact that the uptake of ammonium ions causes an increase in the release of H^+^ by fungal cells, the variation in phosphorus concentration caused a significant drop in pH from 6.0 to 2.0. The biomass concentration data showed that calcium deficiency negatively affected the adaptation of fungal cells to the different phosphorus/pH conditions. Thus, reduced growth and biomass formation at lower phosphorus/pH levels were observed. A large number of studies on a variety of eukaryotic cell types, including fungal cells, reported interactions between changes in pH and calcium cellular signals, where both cytosolic acidification and alkalization caused increases in cytoplasmic calcium for providing ion homeostasis in the cell [34]. Therefore, calcium availability is critical for pH stress tolerance, as was shown in our study. While a significant growth-inhibiting effect of calcium deficiency was recorded at high (Pi4) and reference (Pi1) phosphorus substrate concentrations, resulting in pH 5.0 and 3.0, respectively, an increase in biomass concentration was observed when calcium was absent in the media with a low phosphorus level (Pi0.5) and pH 2.0. Such a twisting effect of calcium deficiency could be explained by the higher lipid accumulation under the Ca0-Pi0.5 condition, meaning that the biomass increase under this condition was due to the higher lipid content and not elevated growth rate.

In addition to pH stress tolerance, it has been reported that calcium ions are involved in lipid and phosphorus metabolism of eukaryotic cells [16,21]. Thus, the synthesis and accumulation of phosphorus in the form of energy storage compounds such as polyphosphates are linked to the storage of cellular calcium. Polyphosphate granules, also known as acidocalcisomes, are membrane-bound evolutionary conserved organelles found in prokaryotic and eukaryotic cells, whose main function is the accumulation of polyphosphate and cations such as calcium, magnesium, zinc, and sodium [35]. Calcium, as well as other cations, functions as a neutralizing agent for negatively charged polyphosphates in the formation of acidocalcisomes. Therefore, calcium availability is an important prerequisite for the formation of polyphosphate granules. In our study, calcium deprivation led to a decrease in the total phosphorus content in Mucoromycota fungi. Some exceptions where the total phosphorus content in calcium-deficient conditions was higher than when calcium was present in the media were recorded for *M. circinelloides* FRR 5020, *M. racemosus*, and *U. vinacea*. This could be explained by the possible involvement of other cations present in the media, such as magnesium and zinc, in polyphosphate accumulation. Furthermore, it was reported that for the cells grown under an alkaline pH 7.5, the activities of a microbial polyphosphate synthesis enzyme—polyphosphate kinase (PPK)—and a polyphosphate hydrolysis enzyme—exopolyphosphatase (PPX)—were approximately equal [36]. In contrast, at slightly acidic pH (5.5), PPK activity increased sixfold, while PPX activity remained unchanged [36]. This elevation in PPK activity could be responsible for the increased intracellular accumulation of polyphosphate at pH 5.5. This observation is in accordance with our study, where the highest polyphosphate accumulation was observed at pH 5.5.

The positive effect of calcium-starved growth media on lipid accumulation was observed for oleaginous algae [18], although, to the authors’ knowledge, there is no study reporting the role of calcium in the lipogenesis of oleaginous Mucoromycota fungi. Recently, we reported the first indication on the influence of calcium ions on lipid accumulation in oleaginous *M. circinelloides* [7]. The aim of this study was to investigate whether calcium displayed some general or strain-specific patterns in lipid accumulation in Mucoromycota fungi. In this study, the lipid-triggering effect of calcium deprivation was remarkably pronounced in all fungi depending on the phosphorus substrate/pH conditions. Interestingly, the absence of calcium in the medium with Pi0.5/pH 2.0 showed a general effect of increased lipid accumulation in all fungi except *M. circinelloides* FRR 5020. Concerning the effect of pH on lipid accumulation in fungal cells, the reference literature indicates that pH variation in the culture medium affects the lipid composition rather than the total lipid content [37]. Thus, it is possible that energy was translocated to lipid synthesis rather than polyphosphate synthesis. The response to pH variations is suspected to be strain- and species-specific. Therefore, the variation in the calcium availability effect on lipid accumulation in Mucoromycota fungi observed in this study could be associated with the strain-specific response to pH changes in the culture media associated with the different levels of phosphorus substrate.

The observation of a higher total lipid content in Mucoromycota fungi under calcium deficiency at a low pH of 2.0 could presumably be explained by the activation of the unfolded protein response (UPR). UPR is known as a signal transduction pathway activated in a response to ER stress. ER stress can be mediated by the extremely low pH of the surrounding environment (for example, culture medium) or calcium deficiency, and it results in the disruption of the ER protein-folding capacity [38]. Disruption of the ER protein-folding capacity leads to the activation of the UPR signaling system for restoring ER homeostasis. Furthermore, activation of the UPR pathways’ modulating lipid metabolism in cells triggers lipogenesis, which leads to higher accumulation of lipids. Based on our results, calcium might have an important function in activating UPR pathways, as a lipid-triggering effect under acidic pH was observed when calcium was removed from the culture medium. In addition to the UPR-based hypothesis, there are two other hypotheses explaining the lipid-triggering effect of calcium deficiency. One is related to the mediation of antilipolytic pathways through a calcium-sensing receptor (CaSR) triggered by the low cellular availability of calcium ions. This results in enhanced lipid accumulation in cells [21]. The second hypothesis is based on the importance of calcium ions in the basal sensitivity of the sterol-sensing mechanism of the sterol response element binding protein (SREBP) pathway [19]. Reduction in the calcium concentration in the ER changes the distribution of intracellular sterol/cholesterol, resulting in the enhancement of SREBP activation and triggering the synthesis of neutral lipids. Currently, there is no clear evidence showing which of the hypotheses are valid for fungal cells, and more in-depth investigation is needed to understand the role of calcium in the lipogenesis of oleaginous fungi. Moreover, it is worth exploring whether there is a link between polyphosphate and lipid accumulation, and whether calcium simultaneously affects both accumulation mechanisms.

In addition to the observations related to the calcium involvement in the accumulation of polyphosphate and lipids in Mucoromycota fungi, several other interesting observations arose in this study. When harvesting and washing the fungal biomass, it was observed that the biomass of *A. rouxii* and the two *M. circinelloides* strains had a yellow coloration, indicating possible high content of carotenoids (Figure 8). The biomass obtained from Ca0-Pi0.5 media showed the highest pigmentation. It is interesting that the two strains of the same species, *M. circinelloides*, showed different metabolic responses. Biomass production and lipid accumulation in the calcium-deficient Pi0.5 condition notably differed for these strains. Furthermore, the strain *M. circinelloides* FRR 5020 showed higher carotenoid production than *M. circinelloides* VI 04473 did (Figure 8). The ability of carotenoid production by *M. circinelloides* FRR 5020 is most likely the cause of the difference in metabolic behavior of this strain compared to *M. circinelloides* VI 04473.

Carotenoid production for *M. circinelloides* has been reported previously [39,40], with the main factors triggering carotenoid production being light, temperature, and aeration [41,42]. To the authors’ knowledge, this is the first indication of the triggering effect of calcium deficiency on carotenoid production. Since assessment of carotenoids was outside the scope of this study, no further analysis on estimating carotenoid content was conducted. Due to the fact that Ca availability influenced several co-products in fungal biomass, namely lipids, polyphosphates, carotenoids, and possibly chitin/chitosan, Ca availability can be used to optimize the co-production potential and, therefore, the economic feasibility of Mucoromycota fungal fermentation.

## 5. Conclusions

The aim of this study was to investigate the effect of calcium availability on lipid and polyphosphate accumulation in Mucoromycota fungi. Calcium availability is important for polyphosphate accumulation, while calcium deficiency could be beneficial for triggering lipid accumulation in Mucoromycota fungi. It can be concluded that calcium is an important nutrient for the regulation of polyphosphate and lipid accumulation in fungal cells. Thus, calcium availability can be used as an important optimization parameter in bioprocesses utilizing Mucoromycota for lipid and polyphosphate accumulation. Furthermore, it should be noted that pH and possibly phosphorus availability play an important role in the involvement of calcium in the regulation of lipid, polyphosphate, and carotenoid accumulation in Mucoromycota fungi. Further investigations are needed to understand the role of calcium availability on carotenoid synthesis in fungal cells.

## Figures and Tables

**Figure 1 jof-07-00300-f001:**
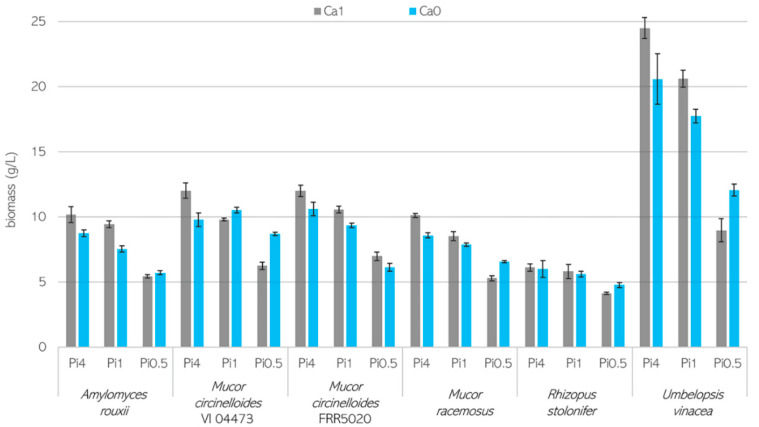
Biomass concentration (g/L) of fungi grown in the presence (Ca1, grey) and absence (Ca0, blue) of calcium under three phosphorus substrate levels (Pi4, Pi1, and Pi0.5). The biomass concentration data represent the average of 4 bio-replicates, and error bars show the standard deviation.

**Figure 2 jof-07-00300-f002:**
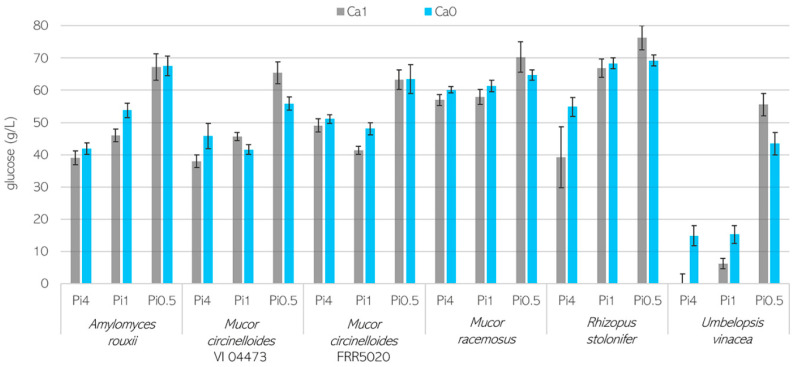
The residual glucose in g/L in the culture supernatants. The glucose concentration was estimated using a partial least square regression model based on the Fourier-transform infrared–attenuated total reflectance (FTIR-ATR) spectra of culture supernatants and reference glucose solutions. Initial glucose concentration was 80 g/L. Average values of four bio-replicates are presented, and error bars represent the standard deviation.

**Figure 3 jof-07-00300-f003:**
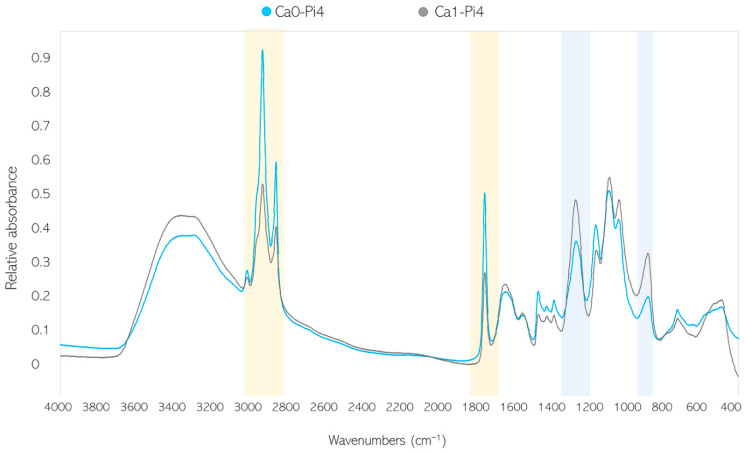
Fourier-transform infrared spectroscopy–high-throughput screening (FTIR-HTS) spectra of *Mucor circinelloides* VI 04473 biomass produced in Ca0-Pi4 (blue) and Ca1-Pi4 (grey) media. The main, characteristic lipid-related peaks are highlighted in yellow, while the polyphosphate-related spectral regions are highlighted in blue.

**Figure 4 jof-07-00300-f004:**
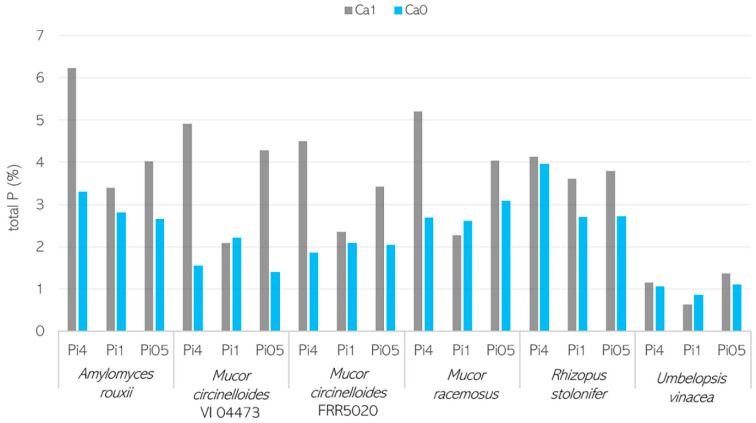
The total cellular P in wt% of dry biomass, estimated using assay-based UV–Vis spectroscopy.

**Figure 5 jof-07-00300-f005:**
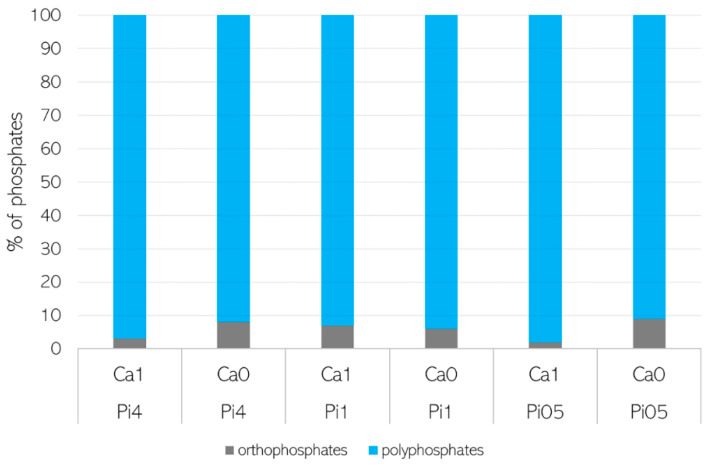
The characterization of cellular phosphates in *Mucor circinelloides* VI 04473 biomass.

**Figure 6 jof-07-00300-f006:**
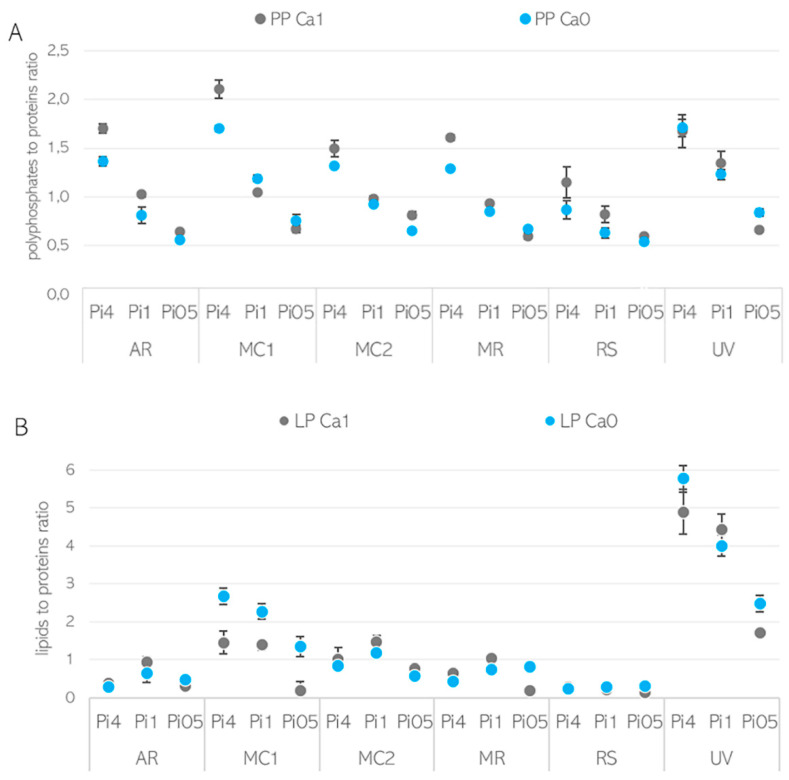
(**A**) Polyphosphate-to-protein (PP) and (**B**) lipid-to-protein (LP) ratios of characteristic bands in FTIR spectra of fungal biomass. Both shown for the presence (Ca1, grey) and absence (Ca0, blue) of calcium in the growth media.

**Figure 7 jof-07-00300-f007:**
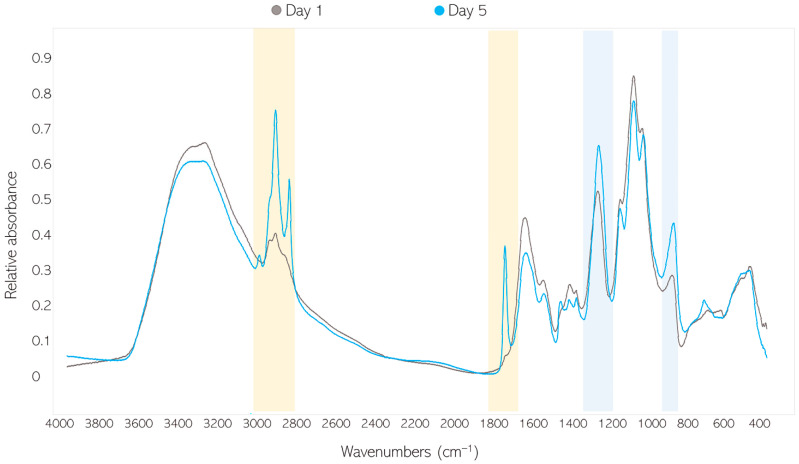
FTIR-HTS spectra of *Mucor circinelloides* VI 04473 biomass grown in Pi4-Ca1 media for 1 (grey) and 5 days (blue). Lipid-related peaks are highlighted in yellow; polyphosphate-related peaks are highlighted in blue.

**Figure 8 jof-07-00300-f008:**
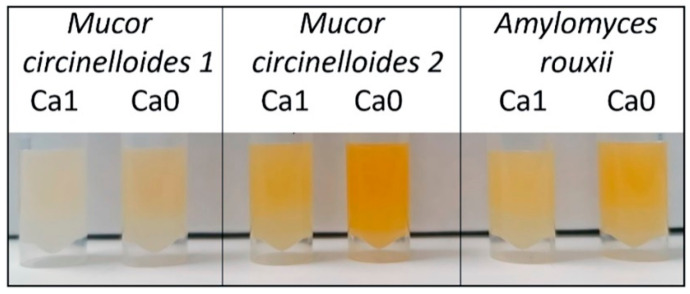
Fungal biomass grown in Pi0.5 media showing carotenoids content. A visibly higher content of carotenoids is observed in biomass grown in Ca-lacking media.

**Table 1 jof-07-00300-t001:** Fungal strains used in the study.

Fungal Strain	Collection №	Short Name
*Amylomyces rouxii*	CCM F220	AR
*Mucor circinelloides*	VI 04473	MC1
*Mucor circinelloides*	FRR 5020	MC2
*Mucor racemosus*	UBOCC A 102007	MR
*Rhizopus stolonifer*	CCM F445	RS
*Umbelopsis vinacea*	CCM F539	UV

**Table 2 jof-07-00300-t002:** Lipid content in % per cell dry weight (d.w.).

Sample		Pi4	Pi1	Pi05
*Amylomyces rouxii*	Ca1	30.37 ± 0.28	46.76 ± 1.80	27.36 ± 0.14
	Ca0	31.24 ± 0.49	40.02 ± 0.32	37.48 ± 1.59
*Mucor circinelloides*	Ca1	42.80 ± 0.00	47.85 ± 0.48	22.67 ± 2.96
VI 04473	Ca0	47.42 ± 1.74	54.01 ± 1.56	48.05 ± 0.36
*Mucor circinelloides*	Ca1	34.37 ± 0.24	47.62 ± 2.48	37.11 ± 0.70
FRR 5020	Ca0	41.01 ± 1.68	48.60 ± 2.07	35.79 ± 0.24
*Mucor racemosus*	Ca1	31.10 ± 0.83	37.85 ± 0.08	22.83 ± 3.42
	Ca0	30.86 ± 3.09	35.04 ± 0.62	39.63 ± 1.82
*Rhizopus stolonifer*	Ca1	25.33 ± 1.06	24.27 ± 0.51	22.78 ± 0.55
	Ca0	27.40 ± 1.70	26.75 ± 1.71	27.90 ± 0.37
*Umbelopsis vinacea*	Ca1	69.90 ± 1.66	81.04 ± 1.93	52.36 ± 3.21
	Ca0	58.43 ± 1.08	84.18 ± 2.94	66.70 ± 0.21

## Data Availability

The data generated in this study are presented in the manuscript and Appendix A.

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
