# Peer review of "Calcium Affects Polyphosphate and Lipid Accumulation in Mucoromycota Fungi"

_jof, 2021, doi:10.3390/jof7040300_

Round 1

Reviewer 1 Report

Dear authors, 

this is very interesting and scientifically very high quolity research, still in some parts it is difficult to read and understand since it looks like several repetitions have been made through the text. It seems that more practice in writing is needed. Anyway this is very good work and relly comprehensive...so just few suggestions have been stressed in this review in order to be more clarified and easier to read and understand.

All the best in further work!  

Author Response

Dear Reviewer, 

Authors are thankful for your valuable comments to the manuscript. Below we address all comments raised in the revision.

Line 69, 71: Citation was modified to Wang et al.

Line 72: Abbreviation ER was used instead of endoplasmic reticulum through the whole manuscript.

Line 78: Sentence ‘As mentioned above, our recent study reported that a variation in the availability of Ca ions affects lipid and polyphosphate accumulation in oleaginous Mucor circinelloides’ was removed from the manuscript text to avoid repeated information.

Line 80: according to the Reviewer’s comment, a short description of oleaginous microorganisms was added in the beginning of introduction (Lines 35-37): ‘Oleaginous Mucoromycota can accumulate lipids with similar composition as plant and animal oils in amounts higher than 20% of their dry cell biomass.’

Line 80: Paragraph from line 80 to 91 was simplified and methodology information was reduced:

 ‘In order to investigate whether the role of calcium ions in lipid and polyphosphate accumulation is conserved for different Mucoromycota fungi, in this study,  six Mucoromycota strains were grown in the presence or absence of Ca ions at three different phosphorus concentrations and thus different pH conditions. Relatively high phosphate concentrations were used to buffer the growth media and provide conditions for polyphosphate accumulation. Nitrogen limitation was used to trigger the lipid accumulation in oleaginous fungi.’

Line 93: Sentence was modified according to Reviewer’s suggestion ‘…, this study is among first assessing the role of Ca…..’

Line 184: changed to ‘fungal samples’

Line 243: All full names of fungal species were abbreviated after first mentioning (e.g. Umbelopsis vinacea- U. vinacea)

Line 382: Figure was abbreviated to Fig. throughout the manuscript.

Line 389: Mucor circinelloides formatted to italic.

Line 414: dry weight was abbreviated to d.w.

Line 462: abbreviations were used M. circinelloides FRR5020, M. racemosus and U. vinacea.

Line 474: To the authors knowledge, there is no study that reports effect of Ca starvation on lipid accumulation in Trichoderma species.

Line 476: modified to ‘we have reported’

Reviewer 2 Report

The study is quite impressive and scientifically elaborated. I advise some corrections:

- Improve the quality of graphics.

- Ensure the uniformity in the units (e.g. mg/mL, µL) throughout the MS.

- Check the references in accordance with the journal style.

Author Response

Dear Reviewer, 

authors are thankful got your valuable comments to the manuscript. Below we address all comments raised in the revision.

The quality of graphics was improved. Resolution is according to journal requirements and deficiencies could have occurred due to conversion of the MS into pdf format.

The uniformity of units was corrected throughout the MS.

References were corrected in accordance with the journal style- journals titles were abbreviated.